# Individualized Goal Setting for Pediatric Intensive Care Unit-Based Rehabilitation Using the Canadian Occupational Performance Measure

**DOI:** 10.3390/children10060985

**Published:** 2023-05-31

**Authors:** Youngsub Hwang, Jeong-Yi Kwon, Joongbum Cho, Jaeyoung Choi

**Affiliations:** 1Department of Health Sciences and Technology, Samsung Advanced Institute for Health Sciences and Technology, Sungkyunkwan University, Seoul 06355, Republic of Korea; 2Department of Physical and Rehabilitation Medicine, Sungkyunkwan University School of Medicine, Samsung Medical Center, Seoul 06351, Republic of Korea; 3Department of Critical Care Medicine, Sungkyunkwan University School of Medicine, Samsung Medical Center, Seoul 06351, Republic of Korea

**Keywords:** cross-sectional study, intensive care unit, occupational therapy, pediatric critical care unit

## Abstract

The Canadian Occupational Performance Measure (COPM) is a client-centered outcome measure that facilitates the prioritization of individualized interventions. Given the rising emphasis on individualized intervention in pediatric intensive care units (PICUs), this cross-sectional study aimed to explore caregivers’ perspectives on their children’s functional goals within PICUs. From 1 September 2020 to 26 June 2022, caregivers of 41 children aged 1–18 years completed the COPM within 48 h of PICU admission. The study also explored the clinical variables predicting a high number of occupational performance goals (≥4/5). Out of 190 goals proposed by caregivers, 87 (45.8%) pertained to occupational performance, while 103 (54.2%) were related to personal factors. Among the occupational performance goals, the majority were associated with functional mobility (55; 28.9%), followed by personal care (29; 15.2%) and quiet recreation (3; 1.6%). Among personal goals, physiological factors (68; 35.8%) were most common, followed by physical factors (35; 18.4%). We found caregiver anxiety, measured by the State-Trait Anxiety Inventory-State, to be a significant predictor of the number of occupational performance goals. These findings underscore the importance of caregiver psychological assessment in the PICU to facilitate personalized goal setting and improve rehabilitation outcomes.

## 1. Introduction

Health systems recognized that morbidities affecting physical and cognitive health and health-related quality of life needed to be addressed [1,2,3,4,5], considering the decline in the mortality of children admitted to pediatric intensive care units (PICUs) [6]. Thus, PICU culture was transformed to facilitate early rehabilitation to minimize morbidities [7], and the emerging literature suggests that the implementation of a structured early mobilization program conducted as part of a multidisciplinary approach in PICUs is feasible and safe [8].

In the early mobilization of PICUs, it is recommended that physical activity be individualized in accordance with each child’s clinical and medical condition, the availability of materials and equipment, age, and the hopes of the family [9]. At this point, the role of occupational therapists (OTs) becomes crucial. OTs are health professionals who help individuals across the lifespan do the things they want and need to do through the therapeutic use of daily activities (occupations). Their holistic and customized approach to evaluations, interventions, and outcomes helps children with disabilities participate in school and social situations, assists people recovering from injuries to regain skills, and provides support to older adults experiencing physical and cognitive changes.

In the context of a PICU, OTs apply the person–environment–occupation performance model [10] to provide client-centered care to achieve rehabilitation objectives within this critical setting. This framework considers the interactions among the person (e.g., children’s goals), environment (e.g., PICUs), and functional task (e.g., supine to sit transfer). In other words, OTs play a central role in facilitating children in the PICU to progress toward or return to crucial daily activities based on each child’s condition. However, while OTs do not exclusively work in PICUs, their services are vital in these settings to address the unique needs of critically ill children. However, to date, evidence regarding the impact of early client-centered and task-specific training performed by OTs within the critical care setting is lacking [11]. Despite growing awareness of the importance of OTs in PICU-based rehabilitation, there is a paucity of evidence on outcomes since PICU occupational assessment tools are lacking.

The Canadian Occupational Performance Measure (COPM) [12] is a client-centered outcome measure designed to identify occupational performance issues during the initial administration of intervention. The COPM serves as a foundation for establishing targeted outcomes and evaluating the effects of treatment [13] and allows for the shaping of a hierarchy within the order of interventions to be conducted. Thus, early administration of the COPM may help in identifying issues in the early phase of rehabilitation. In this individualized and client-centered outcome measure, healthcare professionals not only address diseases but also deeply involve themselves as healthcare professionals caring for the children’s health. Given that client-centered practice has a positive effect on processes and functional outcomes [14,15,16], we hypothesize that the application of the COPM to children in PICUs could significantly enhance both the early rehabilitation process and its outcomes. However, with the diverse functional status of children admitted to PICUs, expecting uniform results for all caregivers is a challenge.

This study has three primary objectives. First, we aimed to understand caregivers’ perspectives on their children’s functional goals in PICUs using the COPM. Second, we sought to assess the feasibility of using the COPM in the early phase of rehabilitation in PICUs, measuring factors such as the time taken to complete the assessment and the ease of presenting a goal list by the caregivers. Lastly, we aimed to identify clinical indicators that could predict a higher number of occupational performance goals, which involved evaluating the correlation between the generated goal lists and key clinical variables.

## 2. Materials and Methods

### 2.1. Study Design and Population

This cross-sectional study was performed from 1 September 2020 to 26 June 2022. The review board of Samsung Medical Center, Seoul, Republic of Korea, approved the present study on 3 August 2020 (Approval No. 2020-05-136). We enrolled children admitted to the PICU of a territory hospital in Seoul, Republic of Korea. The inclusion criteria were (a) 1–18 years of age and (b) admission to the PICU because of critical illness. The exclusion criteria were (a) increased intracranial pressure, (b) history of brain surgery, (c) fracture, (d) admission to the participating PICU ≤ 3 days, and (e) suspected brain death. Informed consent was obtained from the parents or guardians of the children during enrollment. All potential participants underwent a comprehensive face-to-face screening assessment by a PICU specialist. This assessment included a review of the child’s current medical status, the child’s potential for participation in the study, and the feasibility of the intervention within the context of the child’s PICU stay. Subsequently, the parents completed the COPM, which was administered by a single trained OT, within 48 h of PICU admission. We obtained clinical characteristic data from the Samsung Medical Center’s electronic health record, which is used throughout the hospital network.

### 2.2. Caregivers’ Perspectives on Their Children’s Functional Goals

In this study, we utilized the COPM to understand caregivers’ perspectives on their children’s functional goals. We categorized the goals generated by the caregivers into two main factors: occupational performance factors and personal factors.

Occupational performance factors represent the tasks and activities that the children need to, want to, or are expected to perform in their daily lives. In our analysis, we further subdivided these factors into (1) personal care, (2) functional mobility, and (3) quiet recreation.

Personal factors, distinct from health conditions or states, encompass the physiological and physical aspects that caregivers perceive as affecting their children’s functional performance. Accordingly, we subdivided personal factors into (1) physiological factors and (2) physical factors.

It is important to note that while occupational performance factors are directly linked to specific tasks and activities, personal factors serve as underlying elements that could potentially influence the child’s capacity to perform these tasks or activities.

To capture the perceived significance of each goal, caregivers rated its importance on a 10-point scale, where 1 signifies “not important at all” and 10 denotes “extremely important.” This allowed us to gauge the priority each caregiver placed on the respective functional goals and personal factors.

### 2.3. Feasibility Measurements

To demonstrate the feasibility of the COPM at PICU admission, we analyzed the administration process based on the following: (a) how long it took to complete the assessment and (b) how difficult it was for caregivers to present the goal list. The time was recorded in minutes, and caregivers were asked to rate their response on a 10-point scale (1 = extremely easy; 10 = extremely difficult or not able to) about how easy it was to present the goal list of their child.

### 2.4. Clinical Indicators Predicting the Generation of Occupational Performance Goal List

Considering that the COPM was developed with the aim of evaluating clients’ perspectives of performance, we additionally analyzed the number of goal lists generated as occupational performance factors (nOP) among the five most important goals presented by parents (e.g., if three of the five derived goals are occupational performance goals, then the nOP value is 3). Subsequently, we analyzed the correlation between nOP and clinical variables derived at PICU admission to investigate the clinical indicators predicting high nOP. The clinical variables collected were as follows: (a) Pediatric Risk of Mortality III (PRISM III), (b) ventilator use, (c) reasons for PICU admission, (d) Pediatric Evaluation of Disability Inventory Computer Adapted Test (PEDI-CAT), (e) State-Trait Anxiety Inventory-State anxiety (STAI-X-1), (f) previous PICU admission, (g) demographics (age and sex), and (h) Functional Status Scale (FSS).

The PEDI-CAT is a standardized assessment that measures functional activities in children and young people (from birth through 20 years of age) with a variety of health conditions. The PEDI-CAT incorporates a 276-item computer-adaptive platform based on caregiver reporting and includes four domains: mobility, daily activities, social/cognitive function, and responsibility [17,18]. Each domain is scored on a scale from 20 (low function; high caregiver assistance) to 80 (high function; low caregiver assistance). There are two versions of the assessment, the Content-Balanced PEDI-CAT and the Speedy PEDI-CAT, and we used the Speedy version because of its efficiency. We asked parents to respond to the items based on their child’s status one week before PICU admission.

The STAI [19] is a 40-item self-report assessment that measures anxiety and simplifies the separation between trait anxiety and state anxiety. The Korean version of the STAI has been validated. The STAI is divided into the STAI-X-1, evaluating the current state of anxiety, and the STAI-X-2, evaluating aspects of anxiety proneness. In this study, we used the Korean version of the STAI-X-1 [20], which has 20 items. The total score on each subtest was 20–80, with a higher score indicating greater anxiety. We administered STAI-X-1 to the caregivers during the first 48 h of PICU admission.

The PRISM III score is a commonly used mortality prediction model that measures the patient’s most abnormal variables during the first 12 or 24 h in an ICU. The PRISM III score ranges from 0 to 74, with a higher score indicating a higher risk of mortality. The PRISM III score evaluation was executed in accordance with the recommendations of Pollack et al. [21].

The FSS is a measure of functional outcomes and specifically assesses respiratory status, feeding function, motor function, communication, sensory function, and mental status [22]. The total FSS score ranges from 6 (normal function) to 30 (severe dysfunction). We administered the FSS during the first 24 h of PICU admission.

### 2.5. Statistical Analyses

We used SAS 9.4 (SAS Institute Inc., Cary, NC, USA) and R 3.5.0 (R Foundation for Statistical Computing, Vienna, Austria) for all statistical analyses. We used multiple logistic regression models to calculate the odds ratios (OR), 95% confidence intervals (95% CIs), and corresponding *p*-values. A stepwise variable selection method was used in the logistic regression model to evaluate the prediction model. Propensity scores were calculated using the prediction model, which was then used to identify the best cut-off value. The PROC package was used to calculate the area under the receiver operating characteristic (ROC) curve (AUC) and the corresponding 95% CI. The significance level was set at <0.05.

## 3. Results

A total of 157 children were screened for eligibility for participation, and 108 children were excluded. Eight children dropped out because of personal reasons, early discharge, or deteriorating conditions. Finally, 41 children (22 boys and 19 girls; mean age = 8.31 years; standard deviation = 5.87 years) were included in this study (Figure 1). In this figure, ’n’ represents the number of participants.

Table 1 describes the distribution of the study population based on demographic and clinical characteristics. The assessor recorded the time required to administrate the COPM using a chronometer. The average time required for the interview and scoring of the COPM was 15.12 min (range, 10–34 min), and the parents’ mean (SD) rating of the ease of presenting the goal list of their child was 4.49 (2.55).

Table 2 shows the goal lists derived by children’s caregivers. Of the 190 goals the caregivers suggested, 87 corresponded to occupational performance (45.8%), and 103 corresponded to personal factors (54.2%). Among the 87 goals generated for occupational performance, the highest number was found in functional mobility (55; 28.9%), followed by personal care (29; 15.2%) and quiet recreation (3; 1.6%). Among the 103 goals generated for personal factors, the highest number was found for physiological factors (68; 35.8%), followed by physical factors (35; 18.4%). Additionally, Table 2 shows that caregivers of children reported the order of importance as follows: physical factors, physiological factors, functional mobility, personal care, and quiet recreation (for a detailed description of results, see Appendix A).

Table 3 describes the result of the logistic regression analysis of PICU admission factors, determining the nOP as over 4. In the multiple logistic regression model, a stepwise selection method was used to select significant clinical indicators for a high nOP (nOP ≥ 4), and the STAI-X-1 score was a significant indicator (Appendix A). Based on the logistic regression model, we created a prediction function for nOP ≥ 4: *p* = $e^{Q}/1+e^{Q}$, and Q = 5.2961 − 0.1429 × STAI-X-1. This function was then evaluated using an ROC curve, which is a graphical plot illustrating the diagnostic ability of our prediction model at every possible cut-off. The AUC provides a measure of how well the prediction model can distinguish between those with less than four and those with four or more occupational performance goals. A higher AUC means the model has better overall performance.

The *p*-value cut-off of 0.080 refers to the best threshold value of our prediction function (*p*), which balances both sensitivity and specificity for predicting nOP ≥ 4. When the calculated *p* is greater than or equal to this threshold (0.080), we predict that caregivers will identify more than four occupational performance goals for their child. This is not a *p*-value in the sense of hypothesis testing for statistical significance, but rather a threshold value for the prediction model.

In the prediction of nOP ≥ 4, the model performed well, with an AUC of 85.8% (Figure 2). The performance of the model in terms of sensitivity and specificity at different cut-off values is detailed in Appendix A.

## 4. Discussion

This study illuminates the potential benefits and drawbacks of applying the COPM within PICUs. COPM, a client-centered outcome measure, is valuable in eliciting caregivers’ perspectives on their children’s goals in the PICU setting. It promotes the identification and prioritization of occupational performance issues that are most meaningful to caregivers and their children. However, the COPM’s heavy reliance on caregivers’ abilities to identify and articulate their child’s functional goals could be a significant challenge in the high-stress environment of a PICU. Additionally, the subjective nature of COPM might lead to variations in outcomes based on caregivers’ perceptions and understandings of their child’s needs. A potential alternative tool for setting occupational performance goals could be the Goal Attainment Scaling (GAS) [23]. GAS, while also individualized, is more quantitatively oriented, allowing for more precise tracking of goal attainment. However, it may be less sensitive to subtle but meaningful changes in performance, which COPM might capture more effectively with its qualitative approach. A critical comparison between these two tools could illuminate more effective strategies for goal setting within the PICU context. For instance, an integrated approach that combines the strengths of both COPM and GAS could be considered. The COPM’s strength lies in its ability to capture the personal and nuanced goals of the caregivers, while GAS provides a quantitative and objective measure of progress toward goal attainment. This combined approach could offer a comprehensive understanding of the child’s goals and more precise tracking of their progress. However, such an approach would require further validation and consideration of the increased administration complexity. Future research should focus on this comparative evaluation and potentially developing an integrated approach to maximize the strengths and address the limitations of each measure.

We found that more than half of responses related to occupational performance goals were functional mobility goals (55/87 = 63.2%), indicating caregivers recognized functional mobility goals as most important (6.72/10). These findings align with existing evidence highlighting the significance of functioning as an outcome for patients and families [24]. Current reports suggest that 81.5% of children experience functional deterioration while in PICUs [25], underscoring the necessity for interventions targeting functional mobility. However, functional outcomes are seldom primary outcomes in critical care units [26] due to the shortage of research on functional outcomes in PICUs [27] and the heterogeneity of the PICU population. The COPM, with its applicability to all ages and open-ended structure [28], can address these issues. It facilitates individualized goal setting, aligning with the concept of Activity-Based Intervention (ABI) in PICUs. Participation in meaningful occupations not only contributes to a sense of well-being and health [29] but also improves satisfaction among clients and providers [30]. Despite the scarcity of ABI studies in ICUs, our study—the first to address the application of the COPM at PICU admission—provides a foundation for future investigations. Especially in the PICU environment, where prolonged immobilization is common, participation in meaningful occupations in alternative ways might be a more critical component for children. If the COPM is more widely used in PICUs, we expect more studies of ABI to be applied directly to children, which can contribute to a more precise approach to early rehabilitation.

An important finding of the present study was that a high anxiety level in caregivers within the first 48 h of PICU admission negatively affected the generation of high nOP. This suggests that heightened caregiver anxiety may hinder an accurate understanding of the assessment’s intent. Consequently, this underscores the need for psychological assessments to identify vulnerable parents before applying the COPM. Encouraging these individuals to participate in the assessment upon revisiting may improve outcomes. Furthermore, our results underscore the importance of early detection and treatment of predictive factors for parental stress in PICUs. Common stressors include the deprivation of the parenting role, uncertainty over the child’s outcome, communication issues with medical staff, and perceptions of inadequate child care [31]. Evidence suggests that communication interventions can alleviate parental stress in the ICU by providing emotional support, fulfilling informational needs, and fostering a collaborative environment between medical staff and families [32,33,34]. Successful interventions often include proactive communication [35], care conferences that foster collaboration between families and intensive care teams [33], and structured communication processes to discuss and document essential contextual factors for decision making [35]. Given that the COPM encourages parental participation in decision-making and treatment processes, it may also help mitigate parental stressors. Therefore, enhancing focus on the mental status of parents in the PICU environment could facilitate this beneficial cycle.

The primary limitations of our study include the relatively small sample size and limited participant demographic, specifically the inclusion of only Asian individuals. This limitation significantly impacts the generalizability of our findings, especially considering that Asian parents were reported to experience significantly higher stress levels related to their children’s disability characteristics compared with their non-Asian counterparts [36]. Selection bias is another potential limitation, which might influence our study outcomes based on the decision of patients to participate. Our efforts to mitigate this bias by promoting the study and providing educational materials notwithstanding, this limitation remains. Our study’s reliance on self-reported measures, such as the COPM and STAI-X-1, is another limitation, as these measures could be influenced by personal bias or misunderstanding of the questionnaires. Lastly, the cross-sectional design of our study only provides a snapshot of the parents’ anxiety states, thereby restricting our ability to comprehend the long-term psychological trajectory of parents in PICUs. Future research should consider adopting prospective study designs and employing follow-up measurements to gain a more comprehensive understanding of parental anxiety progression over time in PICUs. The inclusion of larger, more diverse populations across different races and cultures is crucial to enhance the generalizability of the findings. Future studies should also consider using multiple assessment methods alongside self-reported measures to mitigate the potential bias inherent in self-reporting. Exploration of other potential predictors of high nOP, apart from anxiety levels, would also be an important area for future research, potentially uncovering more nuances in the caregiver–occupational performance relationship. Lastly, given the emphasis on individualized functional goals in our study, research into the development and implementation of personalized interventions in the PICU setting would be valuable in advancing the field of early rehabilitation.

## 5. Conclusions

In this study, we found that the COPM was a feasible tool for the caregivers of children in the PICU. Functional mobility emerged as the most common goal, followed by personal care and quiet recreation, among the occupational performance goal list. Notably, high nOP could be specifically predicted from the clinical parameter STAI-X-1, obtained within 48 h of PICU admission.

The nOP reflects the caregiver’s engagement in setting rehabilitation goals for their child and their understanding of their child’s needs. Therefore, the nOP’s clinical significance lies in its potential as an indicator of caregiver involvement in the goal-setting process. A high nOP suggests active engagement and a comprehensive understanding of the child’s needs, which can inform a more individualized care plan. On the other hand, a low nOP might indicate high anxiety levels in caregivers, pointing to the need for additional psychological support.

Our findings underscore the necessity of early psychological assessment and intervention in PICUs to alleviate parental stress, which could improve rehabilitation outcomes for their children. This study’s findings contribute to setting goals for early rehabilitation in PICUs. Future studies with larger sample sizes and more diverse populations are warranted to validate and expand on our results.

## Figures and Tables

**Figure 1 children-10-00985-f001:**
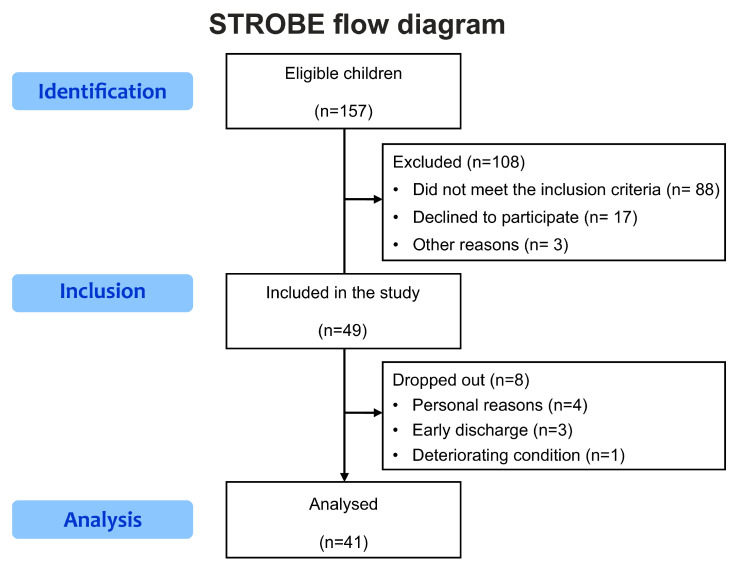
Study enrollment flowchart.

**Figure 2 children-10-00985-f002:**
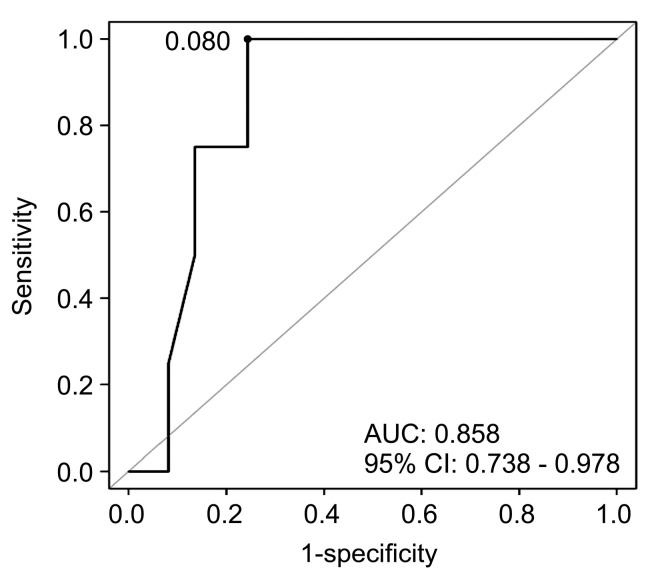
ROC curves and AUC values of the logistic regression model for the analysis of nOP ≥ 3 (1 and 2 vs. 3, 4, and 5); ROC curve and AUC values for the prediction of a high nOP ≥ 4. The prediction equation was *p* = $e^{Q}$/(1 + $$e^{Q}$$), and Q = 5.2961 − 0.1429 × STAI-X-1. The predictability was 85.8%, and the *p*-value cut-off was 0.080. ROC, receiver characteristic curve; AUC, area under the curve; nOP, number of goals generated for occupational performance; CI, confidence interval; STAI-X-1, State-Trait Anxiety Inventory.

**Table 1 children-10-00985-t001:** Demographic and clinical characteristics of children.

Characteristics		n (%)
Age, mean (SD), y		8.31 (5.87)
Sex, n (%)	Boys	22 (53.66)
	Girls	19 (46.34)
STAI-X-1, mean (SD)		57.93 (10.02)
FSS, mean (SD)		11.51 (6.43)
PEDI-CAT, mean (SD)	Daily activities	54.93 (11.12)
	Mobility	60.54 (12.06)
	Cognitive/social	61.39 (12.47)
	Responsibility	44.56 (14.50)
PRISM III, mean (SD)		8.66 (7.55)
Ventilator use, n (%)	Yes	24 (58.54)
	No	17 (41.46)
Previous PICU admission, n (%)	Yes	13 (31.71)
	No	28 (68.29)
Reasons for admission, n (%)	Cardiovascular	6 (14.63)
	Hematology and oncology	7 (17.07)
	Neurologic	8 (19.51)
	Respiratory	16 (39.02)
	Sepsis	4 (9.76)
COPM, mean (SD)	Administration time (min)	15.12 (3.68)
	Difficulty	4.49 (2.55)

STAI-X-1, State-Trait Anxiety Inventory-State anxiety; PRISM, Pediatric Risk of Mortality; PICU, pediatric intensive care unit; SD, standard deviation; n, number; COPM, Canadian Occupational Performance Measure; FSS, Functional Status Scale.

**Table 2 children-10-00985-t002:** COPM goal lists and additional concerns (physiological and physical factors) generated by caregivers.

Goal Domains	n/Total (%)	Importance, Mean (SD)
Occupational performance goal lists		
Functional mobility	55/190 (28.9)	6.76 (1.24)
Actively repositioning in bed	26/55 (47.4)	7.23 (1.19)
Ambulation	11/55 (20.0)	6.54 (1.40)
Sitting whilst leaning back	7/55 (12.7)	6.29 (1.18)
Sitting on the edge of the bed	7/55 (12.7)	6.00 (1.14)
Transfer	2/55 (3.6)	7.00 (0)
Reaching	2/55 (3.6)	6.00 (0)
Personal care	29/190 (15.2)	6.24 (1.42)
Communication	15/29 (51.7)	6.20 (1.52)
Dressing	8/29 (27.6)	6.00 (1.43)
Personal hygiene	6/29 (20.7)	6.33 (1.00)
Quiet recreation	3/190 (1.6)	3.67 (0.44)
Using electronic devices	3/3 (100.0)	3.67 (0.44)
Additional concerns (Personal factors)		
Physiological factors	68/190 (35.8)	7.01 (1.52)
Respiratory function	18/68 (26.6)	8.56 (1.11)
Consciousness	17/68 (25.0)	7.06 (1.25)
Intubation, IV nutrition	17/68 (25.0)	6.11 (1.79)
Urinary catheterization	10/68 (14.8)	6.20 (1.40)
Sensory	3/68 (4.4)	5.67 (0.44)
Body temperature	1/68 (1.4)	7.00 (0)
Cough	1/68 (1.4)	6.00 (0)
Swelling	1/68 (1.4)	7.00 (0)
Physical factors	35/190 (18.4)	7.43 (1.45)
Muscle strength	22/35 (62.8)	7.50 (1.59)
Stiffness	8/35 (22.9)	7.00 (0.50)
Range of motion	5/35 (14.3)	7.80 (0.64)

IV, intravenous; SD, standard deviation; n, number.

**Table 3 children-10-00985-t003:** Logistic regression analysis of pediatric intensive care unit admission factors determining the number of goals generated for occupational performance over 4.

Characteristics	nOP ≥ 4 (0, 1, 2, and 3 vs. 4 and 5)		
	Crude OR	95% CI	*p*-Value
Age	1.011	0.995–1.027	0.195
Sex (females)	0.352	0.033–3.701	0.384
STAI-X-1	0.867	0.756–0.994	0.041 *
PEDI-CAT Daily Activities	1.063	0.970–1.165	0.193
PEDI-CAT Mobility	1.075	0.953–1.213	0.239
PEDI-CAT Cognitive/Social	1.063	0.949–1.189	0.291
PEDI-CAT Responsibility	1.063	0.970–1.165	0.193
PRISM III	0.920	0.772–1.096	0.350
FSS	0.944	0.775–1.149	0.566
Ventilator use, yes	1.467	0.186–11.587	0.716
Previous PICU admission, yes	N/A		
Cardiovascular, yes	8.251	0.898–75.791	0.062
Hematology and oncology, yes	N/A		
Neurologic, yes	N/A		
Sepsis, yes	N/A		
Respiratory, yes	1.643	0.207–13.013	0.638

Crude logistic regression (*p* < 0.05). STAI-X-1, State-Trait Anxiety Inventory; PRISM-III, Pediatric Risk of Mortality III; FSS, Functional Status Scale; PICU, pediatric intensive care unit; * statistically significant value; OR, odds ratio; CI, confidence interval; nOP, number of goal lists generated as occupational performance; N/A, not applicable due to the absence of participants with these characteristics in the nOP ≥ 4 group.

## Data Availability

Data are available on request due to privacy restrictions.

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
