# Peer review of "Individualized Goal Setting for Pediatric Intensive Care Unit-Based Rehabilitation Using the Canadian Occupational Performance Measure"

_children, 2023, doi:10.3390/children10060985_

Round 1

Reviewer 1 Report

In the cross-sectional study “Individualized Goal Setting for Pediatric Intensive Care Unit based Rehabilitation Using the Canadian Occupational Performance Measure”, Youngsub Hwang set up a very interested study with good perspectives for modern PICUs. The sample size is small, and this is a major limitation, acknowledged by the authors, but the messages are of specific importance for those interested in rehabilitation and palliative care starting in the intensive care setting.

I have some minor comments.

41. Occupational therapists (OTs) apply the person–environment–occupation performance model to provide client-centered care to achieve rehabilitation objectives within settings of intensive care units

·      Before this sentence you should describe what, occupational therapists are. In most PICU physiotherapists are working for early rehabilitation and psychologists. I think for those not aware of the differences, you should better describe the role of OTs and if they work exclusively in the PICU.

85. We classified COPM goal lists derived by caregivers into occupational performance factors and personal factors.

·      Please explain what the occupational performance factors and the personal factors stand for. What is the difference between them? 

169. Table 2. The Intubation, IV nutrition and Urinary catheterization are COPM goals. 

·      Were these goals generated by caregivers? Is this correct? You should correct or clarify what does this mean.

178. This function was calculated using a logistic regression model and selection method. The propensity score per the p function and real nOP present were used to calculate the area under the ROC curve (AUC). … In the prediction of nOP ≥4, the predictability was 85.8%, and the prediction p-value cut-off was 0.080 (Figure 2), suggesting that the parents would indicate more than four occupational performance goals in their child.

·      This is difficult for me to follow. I know the ROC analysis and the logistic regression models as separate methods. Here they are mixed up and there are data non understandable by me. For example, what the p-value cut-off was 0.080, stands for? What is 0.080? An insignificant p value or something else? Please make this part of the results section more understandable. 

Author Response

Thank you for your comment. Please see the attachment.

Reviewer 2 Report

I believe that the analyzed topic is of great interest.

However, the study has the following limitations:

- objective of the study expressed in an unclear way

- low population

- poor presentation quality.

Reviewer 3 Report

Thank you for this review. The authors address a new and very timely topic.

Some suggestions for minor revisions:

MATERIALS AND METHODS

Study Design and Population

-       (line 79) All potential participants underwent face-to-face screening assessment…

I think it should be explained what face-to-face screening assessment consists of…Performed by whom? Physicians? Occupational Therapists? Together? In the presence of caregivers/parents?

Caregivers’ Perspectives of Their Children’s Functional Goals

Authors should add references and rationale for this classification, which, sometimes, seems a bit arbitrary. Later, in the results, the division between physiological and physical factors is questionable-is swelling a physiological or physical factor?

Clinical Indicators Predicting the Generation of Occupational Performance Goal List

-       (line 109) f) History of PICU admission

Do you mean previous PICU admission?

-       PEDI-CAT, FSS, PRISM III

For clarity, authors should add the score ranges on each test/domain, similar to what they stated for STAI-X-1.

-       (line 125) We administered STAI-X-1 during the first 48 h of PICU admission.

For clarity, authors should add to whom they administered STAI-X-1. Caregivers/parents, I suppose.

RESULTS

Table 2. COPM goal lists generated by caregivers.

The authors should modify the table columns so that the goal domains will be more visible and differentiated (OCCUPATIONAL PERFORMANCE: Functional mobility - Personal care - Quiet recreation and PERSONAL FACTORS: Physiological factor – Physical factor)  

DISCUSSION

I suggest replacing the first paragraph of the discussion. Let's list the results in the results chapter. The authors would add comments on COPM, its advantages, and its limitations. Are there other tools for setting occupational performance goals? Are comparisons possible?

Reviewer 4 Report

This paper assesses the use of individualised goal-setting by caregivers within the PICU.  It is an interesting paper focusing on the important concept of person-centred decision-making and care.

Introduction:

Generally well-written, however a few sentences require rewording/clarification:

Line 59 - "people who care for the children's health" could mean parents/caregivers, health professionals, or both 

Line 63 - what is "equal level of results"?  Is this common functional outcomes, the same number of outcomes, or something else?  

Methods:

Line 81 – “the test was administered by a trained OT”.  Does this mean only a single OT administered the test every time, or it was only administered by OTs “…administered by trained OTs”.  Would having multiple OTs administer the COPM potentially change the time taken to complete the assessment – is there a standard duration that the assessment normally takes?

Results:

All of the tables are quite difficult to read due to the formatting.  Row lines/shading or similar are needed to separate row groupings on all tables, and the current centre justify for the first column of each table may be better presented if left justified

Table 1: would benefit from row lines/shading to differentiate between categories, “age” appears to be a column header but it isn’t, and “difficulty” is separated from “COPM”.

Table 2: please separate out goal domains and sub-domains as the totals are difficult to follow across domains/sub-domains

Table 3: Left justify for the first column would make this easier to read.  Maybe a note as to why there are “N/A” in table (no nOP1-3 or no nOP4-5?)

What were the characteristics of goals by parent (average, range, etc)?  

Discussion:

I may have missed it, but what is the clinical significance of nOP of 4?  Does it matter if they select 2 or 6 goals?  I understand that the more anxious parents are more likely to select fewer goals, but is this necessarily a negative?  In other areas of health, reducing the number of goals and focusing on a select few is seen as a more productive approach and often results in better outcomes.  Including some literature on this from an OT perspective would be helpful.

Conclusions:

Well-written but would benefit from an extra sentence or two regarding the clinical significance of nOP.
